# Microbial production of multiple short-chain primary amines via retrobiosynthesis

Dong In Kim[1,2,5], Tong Un Chae [1,2,5], Hyun Uk Kim [2,3,4,5], Woo Dae Jang [1,2] & Sang Yup Lee [1,2,4✉]

Bio-based production of many chemicals is not yet possible due to the unknown biosynthetic pathways. Here, we report a strategy combining retrobiosynthesis and precursor selection step to design biosynthetic pathways for multiple short-chain primary amines (SCPAs) that have a wide range of applications in chemical industries. Using direct precursors of 15 target SCPAs determined by the above strategy, *Streptomyces viridifaciens vlmD* encoding valine decarboxylase is examined as a proof-of-concept promiscuous enzyme both in vitro and in vivo for generating SCPAs from their precursors. *Escherichia coli* expressing the heterologous *vlmD* produces 10 SCPAs by feeding their direct precursors. Furthermore, metabolically engineered *E. coli* strains are developed to produce representative SCPAs from glucose, including the one producing 10.67 g L$^{-1}$ of *iso*-butylamine by fed-batch culture. This study presents the strategy of systematically designing biosynthetic pathways for the production of a group of related chemicals as demonstrated by multiple SCPAs as examples.

[1] Metabolic and Biomolecular Engineering National Research Laboratory, Department of Chemical and Biomolecular Engineering, KAIST Institute for BioCentury, Korea Advanced Institute of Science and Technology (KAIST), Daejeon 34141, Republic of Korea. [2] Systems Metabolic Engineering and Systems Healthcare Cross-Generation Collaborative Laboratory, KAIST, Daejeon 34141, Republic of Korea. [3] Systems Biology and Medicine Laboratory, Department of Chemical and Biomolecular Engineering, KAIST, Daejeon 34141, Republic of Korea. [4] KAIST Institute for Artificial Intelligence, BioProcess Engineering Research Center and BioInformatics Research Center, KAIST, Daejeon 34141, Republic of Korea. [5]These authors contributed equally: Dong In Kim, Tong Un Chae, Hyun Uk Kim. ✉email: leesy@kaist.ac.kr

Microbial production of industrial chemicals has garnered much attention as a solution to replace existing petrochemical processes that have caused various environmental problems. To date, the scope of chemicals that are producible using microorganisms continues to expand through advances in systems metabolic engineering[1,2]. Development of a general strategy that allows the construction of a platform strain capable of producing a group of related chemicals can be an efficient approach to expand the portfolio of bio-based products. Such microbial platform strains producing alcohols[3,4], carboxylic acids[4,5], esters[6], and lactams[7,8] have been constructed. In all these studies, promiscuous enzymes that could convert a wide range of substrates into their structurally related chemicals were employed.

Short-chain primary amines (SCPAs) have an alkyl or aryl group in place of a hydrogen atom in ammonia with carbon chain lengths ranging from C1 to C7[9–11] (Supplementary Table 1). SCPAs have a wide range of applications in chemical industries, for example, as a precursor of pharmaceuticals (e.g., antidiabetic and antihypertensive drugs), agrochemicals (e.g., herbicides, fungicides, and insecticides), solvents, and vulcanization accelerators for rubber and plasticizers (Supplementary Table 1). The market size of SCPAs was estimated to be more than 4 trillion US dollars in 2014 (https://www.marketsandmarkets.com/Market-Reports/alkylamines-market-726.html). At present, SCPAs are most commonly produced by dehydrating appropriate alcohols with ammonia using a catalyst under a harsh condition[12]. To the best of our knowledge, despite their industrial importance, biological production of SCPAs has not yet been reported. Only some long-chain primary amines were produced via enzymatic conversions in vitro[13,14]. Thus, it will be of great interest to develop platform microbial strains capable of producing SCPAs from renewable resources.

Microbial production of multiple SCPAs requires careful examination of metabolic pathways, and could be aided by retrobiosynthesis. Retrobiosynthesis allows a systematic design of a biosynthetic pathway by using a set of biochemical reaction rules that describe chemical transformation patterns between substrate and product molecules at an atomic level[15]. An input molecule (i.e., a target chemical) can be back-transformed to a series of intermediate molecules (metabolites), ultimately toward a precursor metabolite; it can be a single step or multiple steps although the former is obviously preferred. Despite the great potential of retrobiosynthesis in systems metabolic engineering, reports on its experimental applications have been rather limited to only a few, including the assimilation of formaldehyde[16], and the production of 1,4-butanediol[17], pinocembrin[18] and 5-aminolevulinic acid[19].

Here, we report the development of platform Escherichia coli strains for the production of various SCPAs. On the basis of their industrial values, we aim at producing 15 different SCPAs, including methylamine, ethylamine, n-propylamine, iso-propylamine, n-butylamine, iso-butylamine, (R)-sec-butylamine, tert-butylamine, n-amylamine, iso-amylamine, 2-methylbutylamine, cyclopentylamine, cyclohexylamine, aniline, and benzylamine, which cover alkyl and aryl amines (Fig. 1a and Supplementary Table 1). We implement retrobiosynthesis (Fig. 1b) and a subsequent precursor selection step (Fig. 1c, d) to design biosynthetic pathways for these SCPAs. We further demonstrate that additional metabolic engineering of E. coli strains allows the enhanced production of representative SCPAs from glucose, including iso-butylamine and ethylamine.

## Results

**Designing biosynthetic pathways of SCPAs.** First, potential biosynthetic pathways of SCPAs were designed using a computational retrobiosynthesis tool we previously developed[20] using a set of reaction rules newly updated in this study ("Methods" and Supplementary Data 1). Retrosynthesis analysis, as a result, showed all 15 different SCPAs might be produced by a single enzymatic conversion from their corresponding direct precursors (Fig. 1b and Supplementary Data 2). For each SCPA, 4 to 21 precursors were predicted, and could be classified, based on their structures, as amino alcohols, amino acids, various forms of SCPAs, alkyl-CoA, and hydrocarbons. These multiple precursors predicted for the possible biosynthesis of each SCPA were sequentially narrowed down for efficient metabolic engineering experiments. For convenience, this procedure of narrowing down precursors will be referred to as the precursor selection step (Fig. 1c, d).

Among the multiple precursors of each SCPA predicted, those classified as amino acids were selected as potential precursors (Fig. 1c and Supplementary Data 2); here, amino acids considered include both proteinogenic and nonproteinogenic amino acids. Because amino acid metabolic pathways are well characterized, and can be efficiently engineered to carry high fluxes[21,22], it was hypothesized that amino acids would serve as ideal precursors of SCPAs. In contrast, metabolic pathways for the biosynthesis of amino alcohols, SCPAs and alkyl-CoA remain poorly characterized, which makes experimental metabolic engineering relatively more difficult. Hydrocarbons were also excluded although their pathways are well known because it was considered more challenging to microbially produce short-chain hydrocarbons[23,24] than amino acids.

According to the retrobiosynthesis predictions, all these amino acid precursors appeared to be convertible by an enzyme with the third-level EC number of 4.1.1.-, which corresponds to the reaction rule #51 (Supplementary Data 1 and 2). Enzymes with the EC number 4.1.1.- are decarboxylases. Since decarboxylation is known to be an irreversible reaction[25], this reaction was expected to serve as a good driving force for strong metabolic fluxes toward SCPA production; previous studies producing n-butanol[26] and biodiesel[27] showed that irreversible reactions contributed to strong metabolic fluxes toward target chemicals. Among various enzymes having the EC number of 4.1.1.-, valine decarboxylase encoded by the vlmD gene from Streptomyces viridifaciens was selected as a proof-of-concept promiscuous enzyme for the production of SCPAs. Valine decarboxylase was previously reported to convert L-valine, L-leucine and L-isoleucine to three different SCPAs, iso-butylamine, iso-amylamine and 2-methylbutylamine, respectively[28] (Fig. 2). Of course, other enzymes having EC number of 4.1.1.- can also be examined. It was hypothesized that valine decarboxylase, if it is promiscuous enough, might be able to convert other amino acid precursors into the corresponding SCPAs of our interest.

Among the 4 to 21 direct precursors predicted for the 15 different SCPAs via retrosynthesis, 1 to 5 direct precursors belonged to a group of amino acids. A single amino acid precursor was further selected for each target SCPA for metabolic engineering experiments by comparing relative positions of the functional groups (i.e., amine and carboxyl groups) in the selected amino acid precursors with L-valine, L-leucine, and L-isoleucine of the VlmD (Figs. 1d, 2); the relative position of a functional group refers to the distance between a functional group and a reaction center carbon atom that is subject to chemical transformation. The rationale behind this criterion is that the feasibility of a chemical reaction is affected by the changes in electron configuration caused by different relative positions of a functional group with respect to a reaction center carbon atom[29]. Furthermore, the relative positions of functional groups play important roles in the binding of a substrate in the active site of an enzyme[30]. As a result, the best amino acid precursor for each of 13 SCPAs was identified (Fig. 2). For tert-butylamine and aniline, there were no potentially good amino acid precursors because they did not have functional groups in the same positions

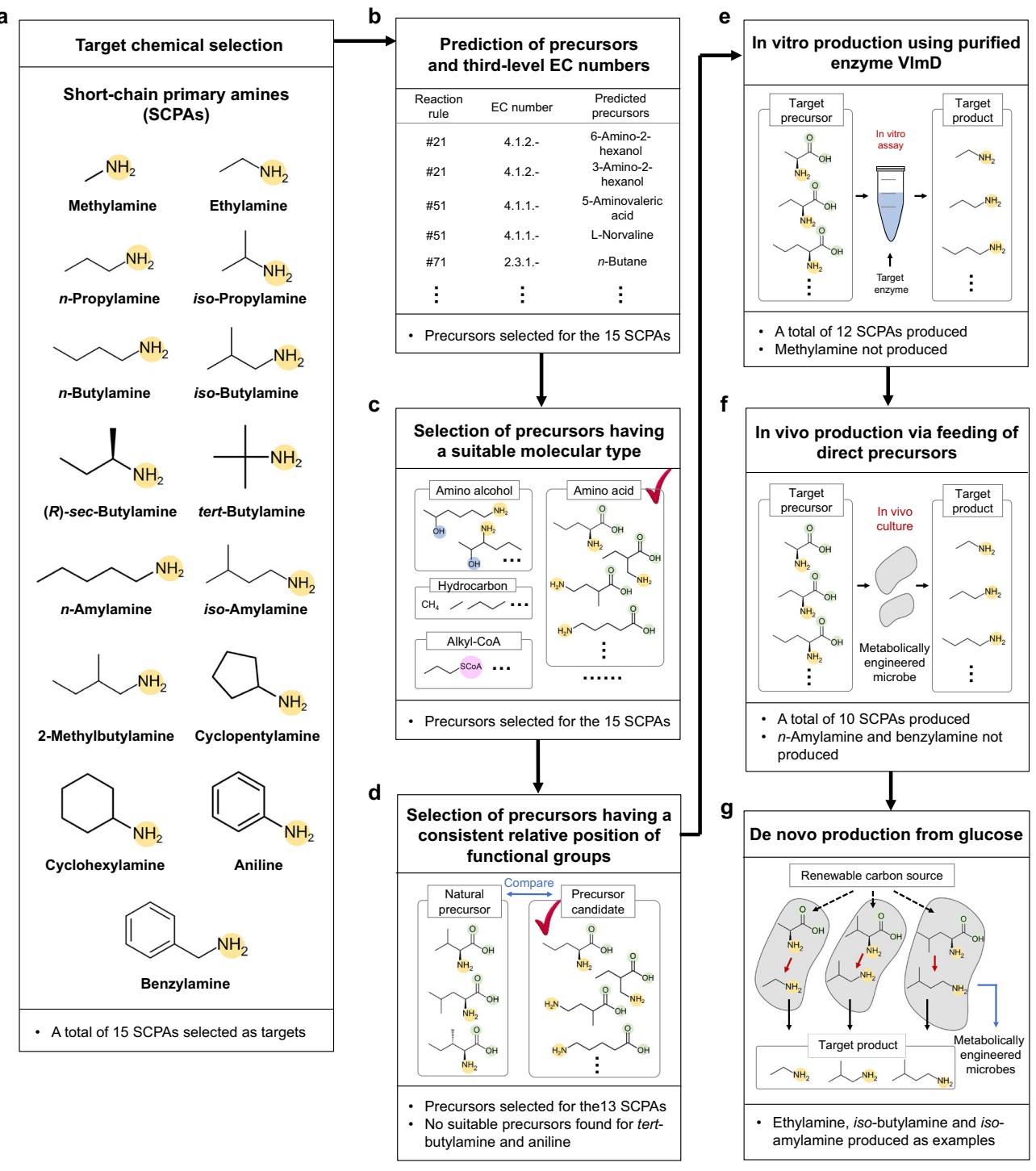

**Fig. 1 Overall workflow for the microbial production of 15 short-chain primary amines (SCPAs).** The workflow implemented in this study largely consists of the following steps: (**a**) target chemical selection; (**b**) prediction of precursors and relevant third-level EC numbers for the generation of these precursors; (**c**) selection of precursors having a suitable molecular type; (**d**) selection of precursors having a consistent relative position of functional groups; (**e**) in vitro production using purified enzyme VlmD; (**f**) in vivo production via feeding of direct precursors; and (**g**) de novo production from glucose. Amine, carboxylic, hydroxy and CoA groups are presented with yellow, green, blue, and pink circles, respectively. In (**c**) and (**d**), these two steps are collectively referred to as the precursor selection step where precursors that meet the selection criteria are presented with a check mark.

as the three VlmD substrates (Fig. 2). Thus, these two SCPAs were no longer considered for production in this study.

**In vitro production of SCPAs.** To validate the results of the retrobiosynthesis and precursor selection step for the production

of 13 different SCPAs from their direct amino acid precursors by VlmD, in vitro enzyme assays were first performed using the purified His$_6$-tagged VlmD (Fig. 1e and Supplementary Fig. 1). VlmD showed in vitro activities on 12 precursors by successfully converting them to their corresponding SCPAs (Fig. 3); one that did not show in vitro activity was the conversion of glycine to

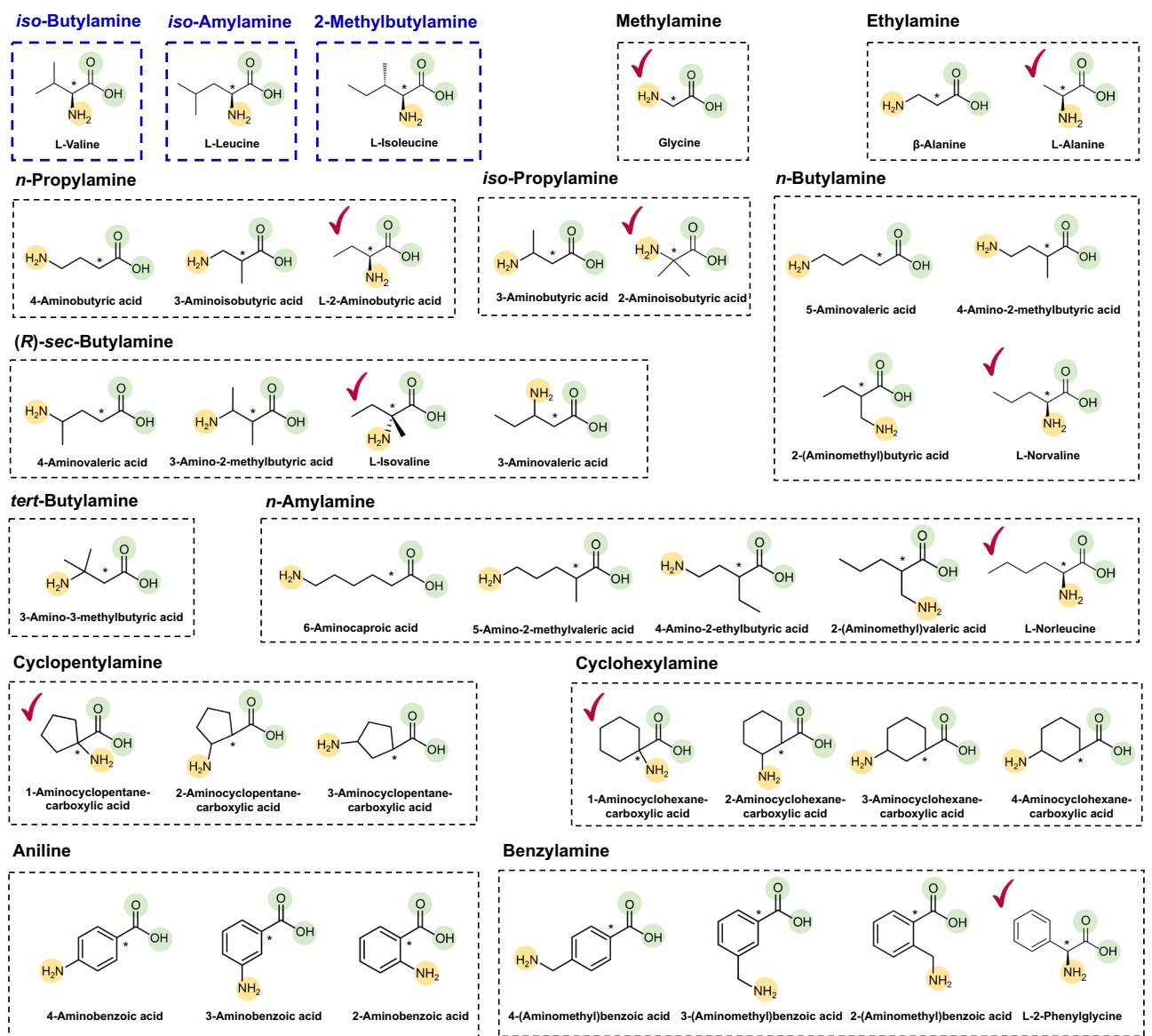

**Fig. 2 Relative position of functional groups in the predicted amino acid precursors of 12 SCPAs in comparison with the previously reported substrates of VlmD.** Amine and carboxylic groups in the amino acid precursors predicted using retrobiosynthesis for each SCPA are presented with yellow and green circles, respectively, in a dotted black box. Reaction center carbon atoms that are subject to chemical transformations are marked with asterisks. Precursors having the same relative position of functional groups, compared with the previously reported VlmD substrates[28] in dotted blue boxes, are presented with a check mark.

methylamine. Also, VlmD consistently showed activities toward L-valine, L-leucine, and L-isoleucine as in the previous study[28]. Among various substrates, VlmD showed relatively high activity toward L-2-aminobutyrate, L-norvaline, L-valine, L-isovaline, L-isoleucine, and 1-aminocyclohexanecarboxylic acid, all showing near or more than 80% conversion to their respective SCPAs: *n*-propylamine, *n*-butylamine, *iso*-butylamine, (*R*)-*sec*-butylamine, 2-methylbutylamine and cyclopentylamine, respectively (Fig. 3). The SCPAs produced from the in vitro enzyme assay reactions were further confirmed by liquid chromatography-mass spectrometry (LC-MS) analysis (Supplementary Fig. 2).

**In vivo production of SCPAs by feeding direct precursors.** To develop platform *E. coli* strains capable of producing 12 different SCPAs in vivo, *E. coli* PA01 was constructed by transforming *E. coli* WL3110 with the plasmid pTac15-vlmD^opt harboring the codon-optimized *S. viridifaciens vlmD* gene (Figs. 1f, 4 and Supplementary Table 2). By individually feeding 2 g L$^{-1}$ of the 12 corresponding precursors, the PA01 strain produced 10 SCPAs in vivo (Fig. 5): 76.28 mg L$^{-1}$ of ethylamine, 873.31 mg L$^{-1}$ of *n*-propylamine, 337.57 mg L$^{-1}$ of *iso*-propylamine, 979.17 mg L$^{-1}$ of *n*-butylamine, 1289.18 mg L$^{-1}$ of *iso*-butylamine, 936.70 mg L$^{-1}$ of (*R*)-*sec*-butylamine, 1052.36 mg L$^{-1}$ of *iso*-amylamine, 1015.75 mg L$^{-1}$ of 2-methylbutylamine, 264.03 mg L$^{-1}$ of cyclopentylamine, and 105.82 mg L$^{-1}$ of cyclohexylamine. However, *n*-amylamine and benzylamine were not produced in vivo, which required L-norleucine and L-2-phenylglycine as precursors, respectively; concentrations of L-norleucine and L-2-phenylglycine in the medium after the cultivation marginally decreased to 1.92 and 1.82 g L$^{-1}$, respectively, from the initial feeding concentration of 2 g L$^{-1}$ (Supplementary Fig. 3), suggesting that these two precursors were poorly utilized by *E. coli*. Both *n*-amylamine and benzylamine should be producible if their corresponding precursors are successfully transported into the

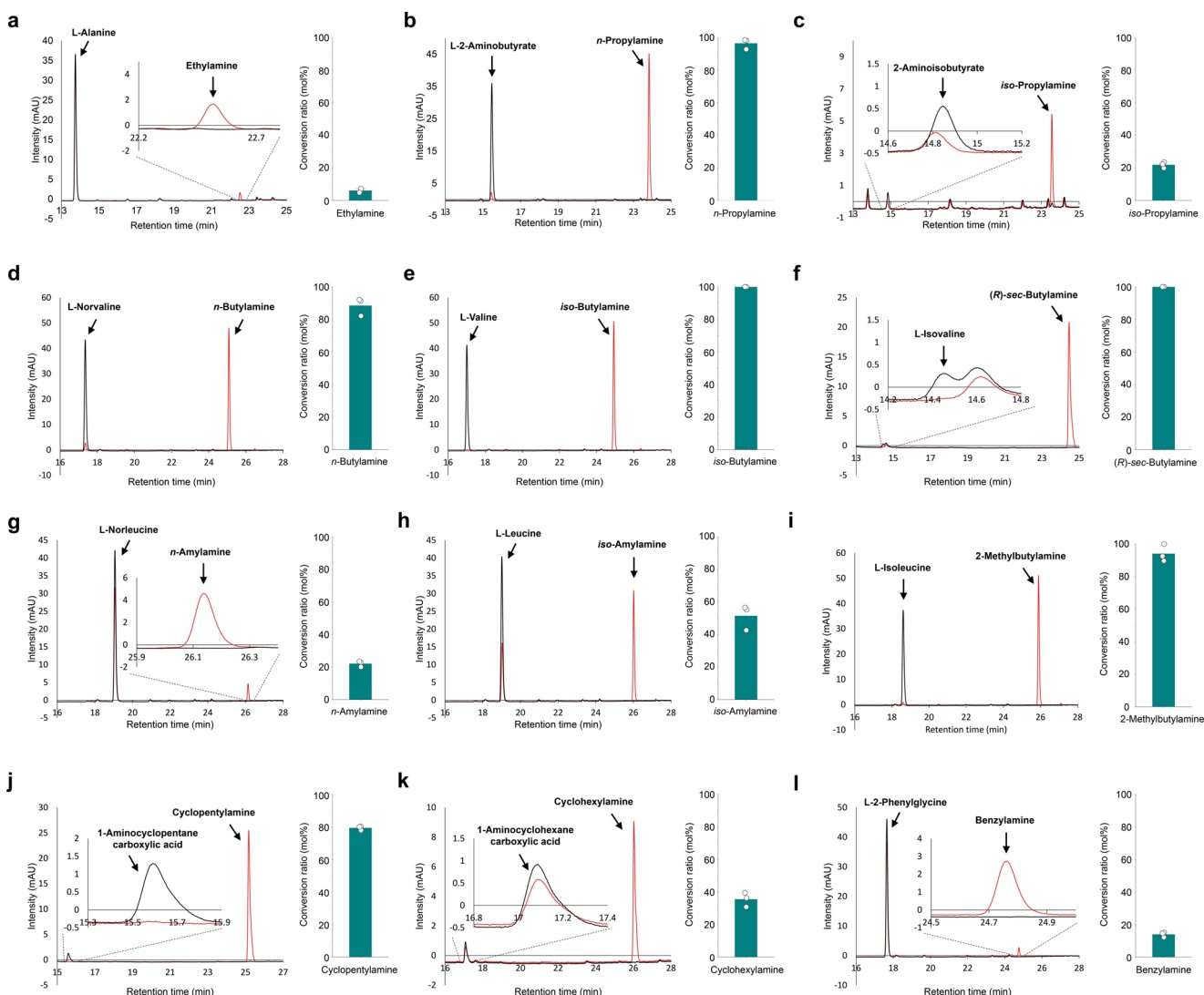

**Fig. 3 In vitro enzyme assay of VlmD producing 12 SCPAs.** High-performance liquid chromatography (HPLC) chromatograms of in vitro enzyme assays conducted using VlmD, which converted: (**a**) L-alanine to ethylamine; (**b**) L-2-aminobutyrate to *n*-propylamine; (**c**) 2-aminoisobutyrate to *iso*-propylamine; (**d**) L-norvaline to *n*-butylamine; (**e**) L-valine to *iso*-butylamine; (**f**) L-isovaline to (*R*)-*sec*-butylamine; (**g**) L-norleucine to *n*-amylamine; (**h**) L-leucine to *iso*-amylamine; (**i**) L-isoleucine to 2-methylbutylamine; (**j**) 1-aminocyclopetanecarboxylic acid to cyclopentylamine; (**k**) 1-aminocyclohexanecarboxylic acid to cyclohexylamine; and (**l**) L-2-phenylglycine to benzylamine. Black and red lines in the left graph for each SCPA indicate chromatograms from in vitro enzyme assays without VlmD (control) and with VlmD, respectively. Green bar graphs indicate percentage (%) of amino acid precursors converted to their corresponding SCPAs by the VlmD after two hours of enzymatic reaction at 37 °C. Experiments were conducted in triplicates. Data are presented as mean values ± s.d. Source data are provided as a Source Data file.

cell, or overproduced from other carbon sources because VlmD was shown to be active on these precursors according to the in vitro enzyme assays (Fig. 3).

**De novo production of SCPAs from a renewable carbon source.** Having confirmed the possibility of microbial production of SCPAs, we next constructed engineered strains for the production of SCPAs from glucose as a sole carbon source and without the external feeding of the corresponding amino acid precursors (Fig. 1g). We selected ethylamine, *iso*-butylamine and *iso*-amylamine as proof-of-concept target SCPAs for the de novo production. First, for the de novo biosynthesis of ethylamine from glucose, alanine dehydrogenase, which converts pyruvate to L-alanine, was introduced to *E. coli* PA01 by overexpressing the *Geobacillus stearothermophilus alaD* gene[31], which was also codon-optimized for the heterologous expression in *E. coli*. The resulting *E. coli* EA01 strain simultaneously overexpressing the

codon-optimized *vlmD* and *alaD* genes was able to produce 2.89 mg L$^{-1}$ of ethylamine from glucose (Fig. 6a and Supplementary Table 2). It should be noted that the EA01 strain secreted 22.78 mg L$^{-1}$ of L-alanine as a byproduct, which was greater than the titer of ethylamine (Fig. 6a). This incomplete conversion seems to have been caused by the insufficient activity of VlmD toward L-alanine, which did not convert all the L-alanine molecules toward ethylamine. Even after employing an engineered VlmD having higher activity for this reaction, it will be important to balance the metabolic fluxes between the generation of a precursor (e.g., L-alanine) and the conversion of a precursor into its product (e.g., ethylamine).

For the production of *iso*-butylamine from glucose, the *E. coli* iBA01 strain was constructed by transforming the L-valine-over-producing *E. coli* Val (pKBRilvBNCED) strain[21], which we previously developed, with pTac15-vlmD$^{opt}$ for the additional expression of the codon-optimized *vlmD* gene. The resulting iBA01 strain produced 1155.74 mg L$^{-1}$ of *iso*-butylamine from

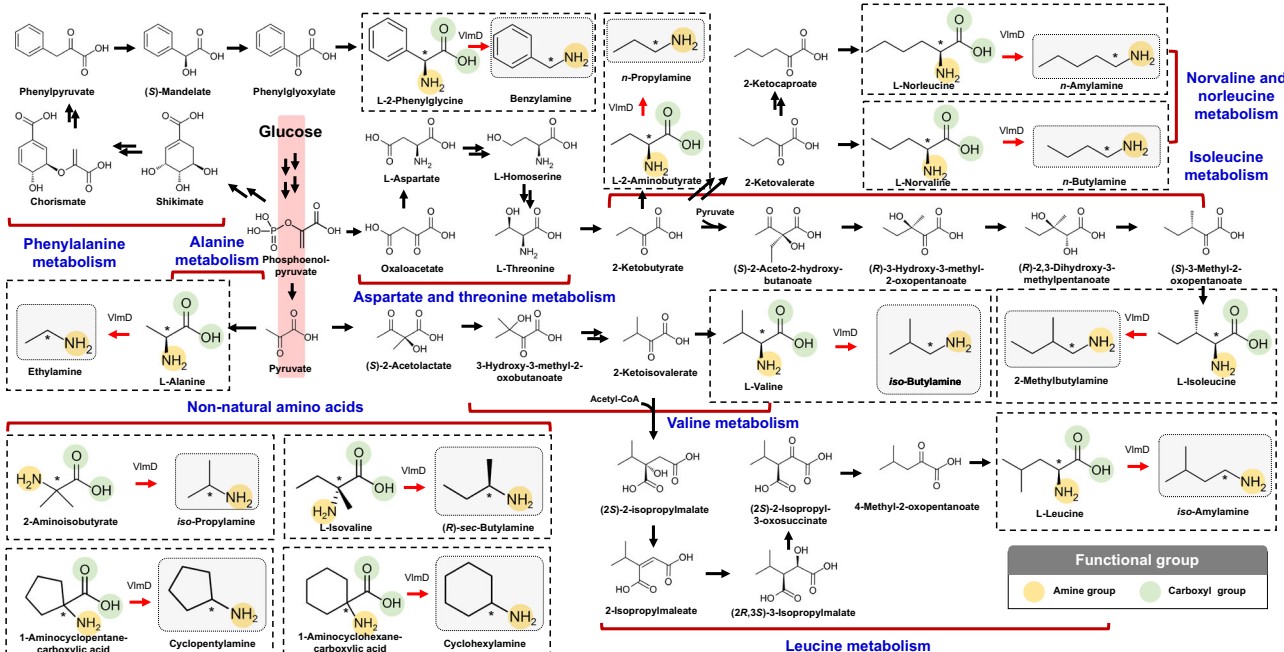

**Fig. 4 Biosynthetic reactions constructed in *E. coli* for the in vivo production of 12 SCPAs.** These 12 SCPAs were the ones shown to be produced by VlmD in vitro (dotted boxes). Amine and carboxylic groups shown in each dotted box are presented with yellow and green circles, respectively. Reaction center carbon atoms that are subject to chemical transformations are marked with asterisks. Glycolysis is indicated with a red background, which leads to the biosynthesis of 12 amino acid precursors. Multiple reactions are presented with two or more arrows.

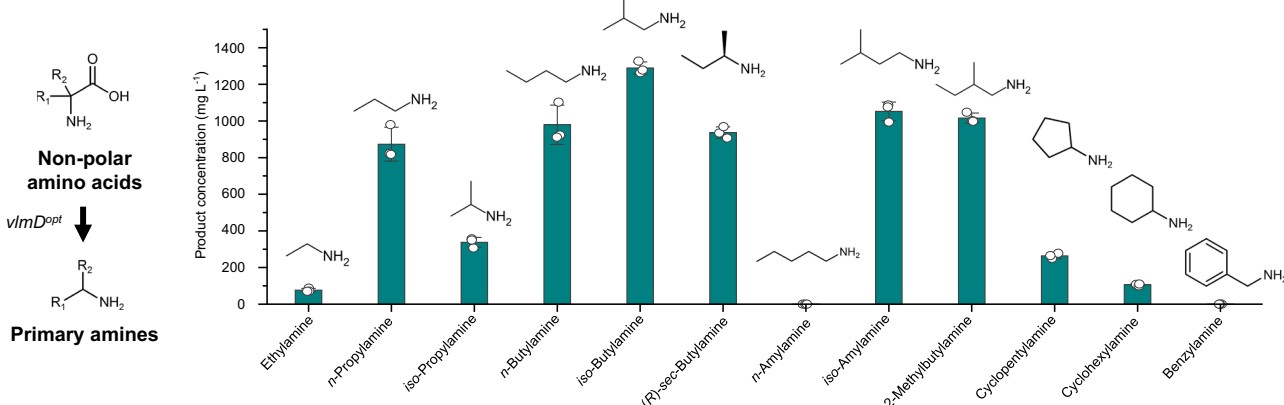

**Fig. 5 In vivo production of 12 SCPAs via feeding of their direct precursors.** Production titers of 12 SCPAs from *E. coli* expressing the codon-optimized *Streptomyces viridifaciens vlmD* (PA01 strain) are presented where direct precursors of the SCPAs were individually provided in the medium. All the experiments for the production of each SCPA were conducted in triplicates. Data are presented as mean values ± s.d. Source data are provided as a Source Data file.

glucose (Fig. 6b and Supplementary Table 2). In contrast to the EA01 strain, the iBA01 strain did not produce L-valine as a byproduct (Fig. 6b); the production titer of *iso*-butylamine (1155.74 mg L$^{-1}$; equivalent to 15.80 mM) obtained with the iBA01 strain was even greater than that of L-valine (524.91 mg L$^{-1}$; equivalent to 4.48 mM) obtained with the NC02 strain, a control strain which is the Val (pKBRilvBNCED) strain harboring an empty plasmid without the *vlmD* gene (Fig. 6b and Supplementary Table 2). This observation suggests that VlmD has a strong driving force toward the *iso*-butylamine production. However, a critical bioprocess problem that needs to be addressed for this strain is that the Val strain is auxotrophic to L-isoleucine, L-leucine, and D-pantothenate because the corresponding biosynthetic genes, *ilvA*, *leuA* and *panB*, respectively, were deleted to remove the competing pathways during the course of strain optimization[21]; we tackle this problem below.

Finally, for the production of *iso*-amylamine from glucose, *leuABCD* genes, all involved in the L-leucine biosynthesis, were additionally overexpressed in the iBA01 strain to increase the precursor, L-leucine, resulting in the iAA01 strain. The iAA01 strain produced 93.23 mg L$^{-1}$ of *iso*-amylamine from glucose (Fig. 6c and Supplementary Table 2). Along with *iso*-amylamine, 991.23 and 759.89 mg L$^{-1}$ of L-leucine and *iso*-butylamine, respectively, were produced as byproducts (Fig. 6c and Supplementary Fig. 4). Production of L-leucine was likely attributed to the suboptimal conversion of L-leucine to *iso*-amylamine by VlmD. Meanwhile, accumulation of *iso*-butylamine was due to the preferred activity of VlmD toward L-valine, which might have been generated from 2-ketoisovalerate, despite the overexpression of the *leuABCD* genes. Indeed, the NC03 strain, a L-leucine-producing strain without the *vlmD*

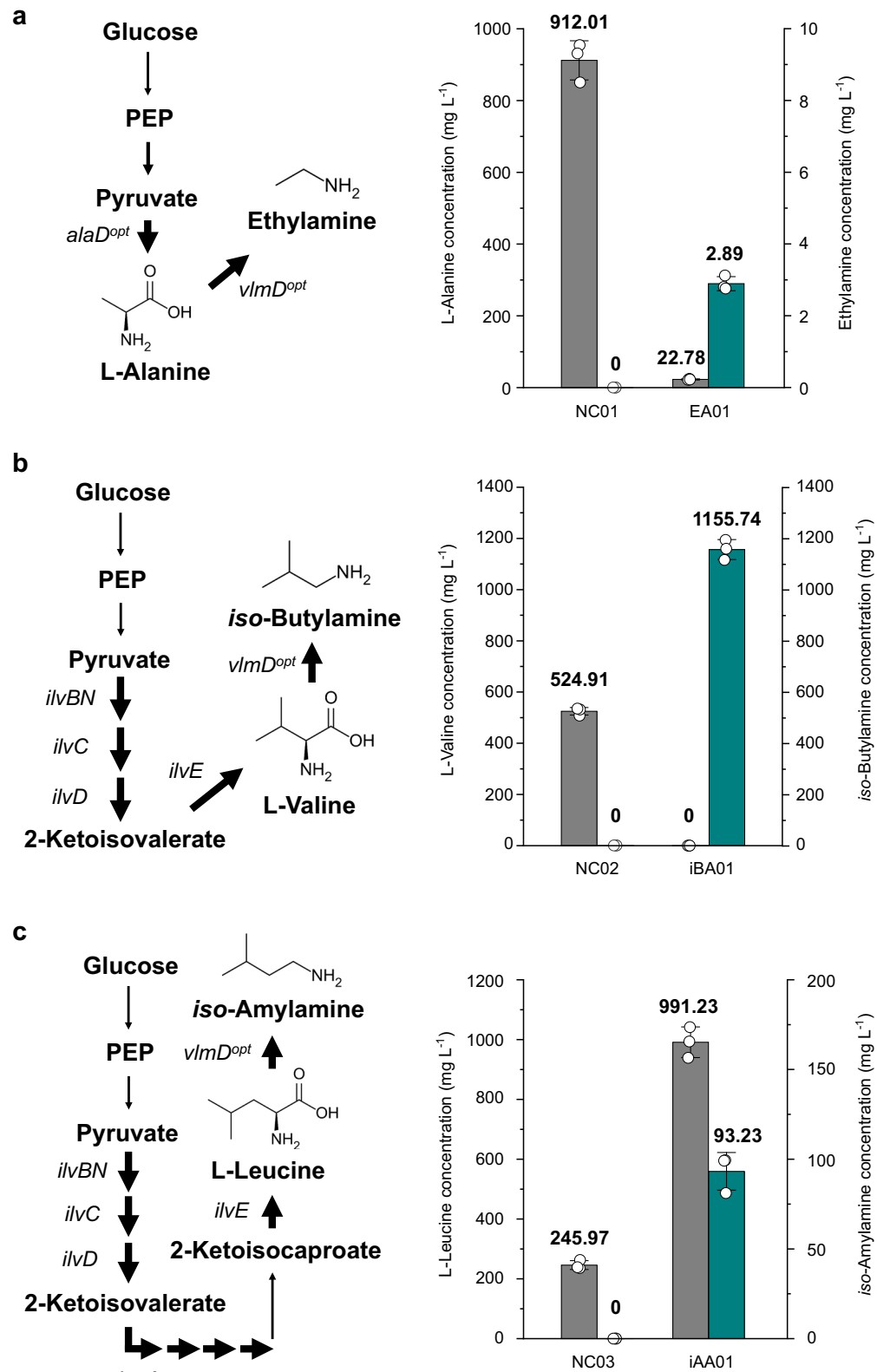

gene, produced 892.67 mg L$^{-1}$ of L-valine as a byproduct (Supplementary Fig. 4). Taken together, the successful de novo production of three representative SCPAs suggests that it will also be possible to biosynthesize other SCPAs using the platform *E. coli* strains from renewable resources through additional relevant metabolic engineering. A next challenge is to overcome the problems that were identified from the de novo production studies, including the iBA01 strain's auxotrophies and the suboptimal activity of VlmD toward L-alanine in the EA01 strain.

**Fig. 6 De novo production of three SCPAs from glucose as a sole carbon source.** Ethylamine, *iso*-butylamine, and *iso*-amylamine were produced from glucose through additional engineering of *E. coli* (Supplementary Table 2): (**a**) ethylamine from the NC01 (control) and EA01 strains; (**b**) *iso*-butylamine from the NC02 (control) and iBA01 strains; and (**c**) *iso*-amylamine from the NC03 (control) and iAA01 strains. The concentrations of amino acid precursors and their corresponding SCPAs are presented with gray and green bars, respectively. Amino acid precursors for the gray bars in (**a**, **b**, and **c**) correspond to L-alanine, L-valine, and L-leucine, respectively. Overexpressed genes are indicated with corresponding gene names. Amplified reactions are shown with thick arrows. Presented genes encode the following enzymes: *alaD*, alanine dehydrogenase; *ilvB*, acetohydroxy acid synthase I large subunit; *ilvC*, ketol-acid reductoisomerase; *ilvD*, dihydroxy-acid dehydratase; *ilvE*, branched-chain-amino-acid aminotransferase; *ilvN*, acetohydroxy acid synthase I small subunit; *leuA*, 2-isopropylmalate synthase; *leuB*, 3-isopropylmalate dehydrogenase; *leuC*, 3-isopropylmalate dehydratase large subunit; *leuD*, 3-isopropylmalate dehydratase small subunit; and *vlmD*, valine decarboxylase. The *alaD*^opt and *vlmD*^opt genes indicate codon-optimized *alaD* and *vlmD* genes, respectively. All the experiments for the SCPA productions were conducted in triplicates. Data are presented as mean values ± s.d. Source data are provided as a Source Data file.

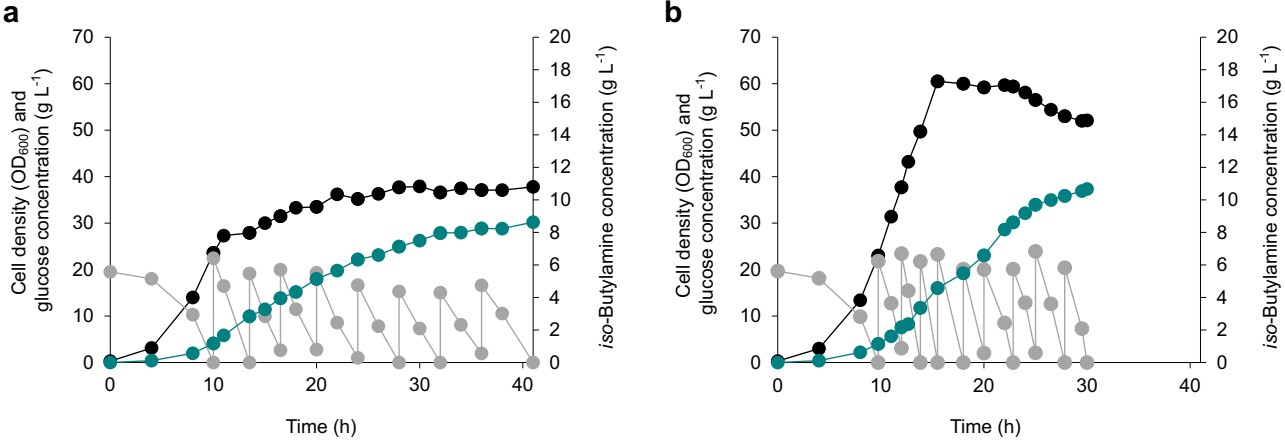

**Fig. 7 Fed-batch fermentations for the enhanced production of *iso*-butylamine. a** Fermentation profile of the iBA01 strain, which was cultured in the NM3 medium that includes L-isoleucine, L-leucine, and D-pantothenate. **b** Fermentation profile of the iBA02 strain, which was cultured in the same NM3 medium as the iBA01 strain, but without L-isoleucine, L-leucine, and D-pantothenate. Another fed-batch fermentation profile of the iBA02 strain, which was conducted for the reproducibility, is available in Supplementary Fig. 5. Cell density (OD₆₀₀), glucose concentration, and *iso*-butylamine concentration are presented with black, gray, and green circles, respectively. Source data are provided as a Source Data file.

**Enhanced production of *iso*-butylamine.** Because the iBA01 strain showed efficient *iso*-butylamine production without forming L-valine as a byproduct in flask cultivation (Fig. 6a), this strain was subjected to fed-batch fermentation using NM3 medium (see Methods for details). As a result, 8.63 g L$^{-1}$ of *iso*-butylamine was produced with a productivity of 0.21 g L$^{-1}$ h$^{-1}$ and a yield of 0.062 g g$^{-1}$ (Fig. 7a). This production titer corresponds to 6.50-fold increase, compared to the titer of 1.15 g L$^{-1}$ obtained in flask culture.

As noted above, however, the iBA01 strain is auxotrophic for L-isoleucine, L-leucine, and D-pantothenate, which poses a problem in industrial fermentation. To solve this problem, instead of knocking out the *ilvA*, *leuA*, and *panB* genes as in Val (pKBRilvBNCED) strain[21], we knocked down the expression of these three genes by employing synthetic small regulatory RNAs (sRNAs)[32]. As a result, the *E. coli* iBA02 strain was constructed by transforming the pTac15-vlmD^opt-anti-ilvA-leuA-panB into the WLGBH (pKBRilvBNCED) strain (Supplementary Table 2). The WLGBH strain is a parental strain of the Val strain, having the intact *ilvA*, *leuA*, and *panB* genes. Fed-batch fermentation of the iBA02 strain showed the maximum OD₆₀₀ of 60.5 without the supplementation of L-isoleucine, L-leucine, and D-pantothenate (Fig. 7b); this value is 59.6% higher than the maximum cell density (OD₆₀₀ of 37.9) obtained with the iBA01 strain with L-isoleucine, L-leucine, and D-pantothenate supplementation. More importantly, the iBA02 strain was able to produce 10.67 g L$^{-1}$ of *iso*-butylamine with a productivity of 0.36 g L$^{-1}$ h$^{-1}$, without the amino acid and vitamin supplementation, although the yield was slightly decreased to 0.057 g g$^{-1}$ (Fig. 7b). These proof-of-concept

experiments suggest that the SCPA-producing platform strain can be further metabolically engineered to overproduce SCPAs of interest.

**Enzyme engineering for the enhanced production of ethylamine.** Among the 12 different SCPAs successfully produced in vitro, VlmD showed relatively low efficiencies in converting L-alanine, 2-aminoisobutyrate, L-norleucine, and L-2-phenylalanine (Fig. 3). To see if this problem can be solved, rational protein engineering of VlmD was conducted to improve its activity toward L-alanine as a proof-of-concept target. L-Alanine was chosen because VlmD showed the lowest in vitro L-alanine conversion efficiency, while the resulting product ethylamine is one of the most widely used SCPAs[12] (Fig. 3). VlmD was first subjected to the homology modeling and docking simulations. Because a 3D crystal structure of VlmD was not available, a homology model of VlmD was generated using glutamic acid decarboxylase (GAD67) from *Homo sapiens* (PDB ID: 2OKJ)[33] (Methods). Since VlmD requires pyridoxal 5′-phosphate (PLP) as a cofactor to catalyze the decarboxylation reaction, PLP-L-alanine was considered in the docking simulation (Fig. 8a). The binding mode of PLP-L-alanine in the homology model of VlmD was observed to be similar to the co-crystal structure of GAD67, which involves the binding of an original ligand PLP-γ-aminobutyric acid (PLP-GABA) (Supplementary Fig. 6); the similar binding modes indicate that residues in the active site of GAD67, which interact with PLP-GABA, can also be considered for the homology model of VlmD. Six residues in the active site of

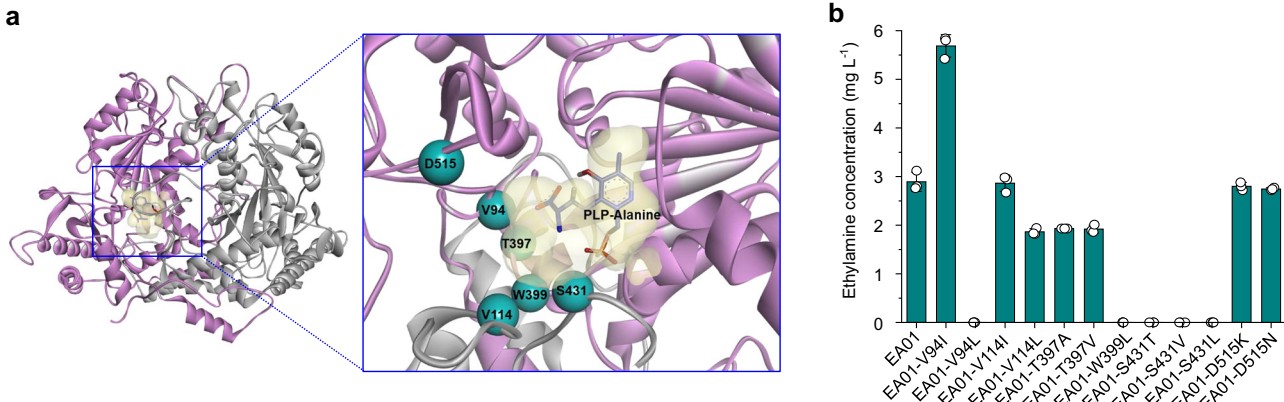

**Fig. 8 Protein engineering of VlmD for the enhanced production of ethylamine. a** Binding model of PLP-L-alanine in the homology model of VlmD. The homology model was generated using glutamate decarboxylase from *Homo sapiens* as a template. Potential six target residues for enzyme engineering are shown in cyan. **b** The concentrations of ethylamine produced by flask cultures of the EA01 strain and its derivative strains having one of the 12 different VlmD variants (Supplementary Table 2). Experiments were conducted in triplicates. Data are presented as mean values ± s.d. Source data are provided as a Source Data file.

the VlmD homology model appeared to be interacting with L-alanine of PLP-L-alanine. Thus, these six residues were selected as target residues for mutations. For their candidate substitutes, amino acids having molecular sizes greater than the corresponding native six target residues were considered because L-alanine is smaller than the preferred substrates (i.e., L-valine, L-leucine, and L-isoleucine) of VlmD[28]. Having larger residues in the active site will reduce the active site volume, thereby increasing the chance of interaction between the active site and L-alanine of the incoming ligand (Fig. 8a). As detailed in "Methods", the following 12 single mutations were suggested for the six target residues: V94I, V94L, V114I, V114L, T397A, T397V, W399L, S431T, S431V, S431L, D515K, and D515N.

To examine the effects of VlmD variants on ethylamine production, 12 engineered *E. coli* strains were constructed by overexpressing each *vlmD* variant and the *alaD* gene, both codon-optimized, in the WL3110 strain (Supplementary Table 2). Flask cultivation of the 12 resulting engineered strains showed that the EA01-V94I strain equipped with the V94I mutant of VlmD produced 5.69 mg L$^{-1}$ of ethylamine from glucose. Although still low, this titer is 96.9% higher than that (2.89 mg L$^{-1}$) obtained with the EA01 strain harboring the native VlmD (Fig. 8b). This result suggests that the activity of VlmD toward other substrates, especially those with a poor conversion, can also be improved via rational protein engineering for the enhanced SCPA production.

## Discussion

In this study, we developed the platform *E. coli* strains for the bio-based production of diverse SCPAs. The application of the combined retrobiosynthesis and precursor selection step allowed the identification of the best direct precursor for each SCPA to be produced and also the third-level EC numbers of candidate enzymes catalyzing the reaction of interest. After confirming the successful in vitro conversion of the direct precursors to SCPAs and in vivo SCPA production by the precursor feeding, the platform *E. coli* strains were further engineered to produce the direct amino acid precursors from glucose for the de novo production of SCPAs. These engineered *E. coli* strains expressing the heterologous valine decarboxylase gene (*vlmD*) were able to produce three proof-of-concept SCPAs from glucose, demonstrating the possibility of bio-based production of diverse SCPAs from renewable resources. To the best of our knowledge, this is the first study on the bio-based production of SCPAs.

Further metabolic engineering was performed to increase the precursor supply and the key enzyme activity (e.g., VlmD in this study) toward the enhanced production of *iso*-butylamine and ethylamine as examples, respectively. In the case of *iso*-butylamine, gene knockdowns via synthetic sRNA technology for the removal of auxotrophies together with fermentation optimization indeed further increased the titer and productivity. Systems metabolic engineering strategies established for amino acids[21,22] will be useful to further boost the production of SCPAs. For ethylamine, the enzymatic activity of VlmD was enhanced by rational protein engineering that involved the implementation of homology modeling and docking simulation. Expressing a VlmD mutant (V94I) suggested from this strategy led to the improved ethylamine production. Although developing a strategy combining retrobiosynthesis and precursor selection step to design biosynthetic pathways was the main objective of this study, the results of the proof-of-concept metabolic and enzyme engineering studies described above suggest that the production of SCPAs can be further enhanced by additional engineering together with fermentation optimization.

Taken together, we developed a strategy combining retrobiosynthesis for the identification of possible precursor metabolites and the best promiscuous enzyme (or at least enzyme candidates), the precursor selection step to determine the best precursor molecules (i.e., based on molecular type and relative position of functional groups) with the best enzyme identified, confirmation of in vitro and in vivo activities, and construction of metabolically engineered strains for the de novo production of SCPAs from glucose. This strategy will be useful for the production of other chemicals, especially a group of related chemicals.

## Methods

**Retrobiosynthesis for producing multiple SCPAs.** A previously developed in-house computational retrobiosynthesis tool[20] was used to generate all possible precursors of target SCPAs. To run this tool, 84 reaction rules were constructed by manually extracting meaningful chemical transformation patterns on the basis of the third-level enzyme commission (EC) numbers from all the biochemical reactions available at the KEGG database[34]. These reaction rules are described in SMiles ARbitrary Target Specification (SMARTS), a chemical language describing structural patterns of molecules (Supplementary Data 1). While our in-house tool was used in this study, retrobiosynthesis using these updated reaction rules mentioned above can also be conducted using the popular ChemAxon Reactor[35] (JChem 19.9.0, 2019; http://www.chemaxon.com/) to generate precursors of a given target chemical. ChemAxon Reactor takes a set of reaction rules and a target

chemical as inputs, which are presented in SMARTS and simplified molecular-input line-entry system (SMILES), respectively. The predicted precursors were exported in the SMILES format. In this study, all possible precursors of the target 15 SCPAs were generated using the in-house retrobiosynthesis tool products by using the 84 reaction rules.

**Precursor selection step**. The molecular types of the retrobiosynthesis-generated precursors were first examined by retrieving information in the 'MeSH Tree' of each precursor available at PubChem[36]. This information was accessed via Pub-ChemPy (v1.0.4). Upon selection of the precursors having the molecular type of amino acid, the relative position of their functional groups was examined next, in comparison with the previously reported substrates (i.e., L-valine, L-leucine, and L-isoleucine) of VlmD. Reaction decoder tool (RDT) (v1.5.1)[37] was used for this purpose. These two steps are implemented in our source code at https://bitbucket.org/kaistsystemsbiology/retro-precursor-selection.

**Enzyme purification and in vitro enzyme assay**. E. coli BL21(DE3) harboring pET-his$_6$-vlmD$^{opt}$ was cultured for the overexpression of N-terminus His$_6$-tagged VlmD in 200 mL of LB medium at 37 °C. The expression of his$_6$-tagged and codon-optimized vlmD gene was induced by adding 1 mM of isopropyl β-D-1-thiogalactopyranoside (IPTG) after 3 h. Cells were collected after additional 5 h of cultivation by centrifugation at 2090 × g for 10 min. Harvested cells were suspended in 50 mL of equilibrium buffer that consists of 50 mM Na$_3$PO$_4$ and 300 mM NaCl (pH 7.0 by HCl). The cells were disrupted using ultrasonic homogenizer (VCX-600; Sonics and Materials Inc., Newtown, CT) with a titanium probe 40 T (Sonics and Materials Inc.). Cell debris was separated by centrifugation at 15,044 × g and 4 °C for 10 min, and the resulting supernatants were loaded onto Talon metal affinity resin (Clontech, Mountain View, CA). Equilibrium buffer supplemented with 7.5 mM of imidazole (10 mL) was subsequently flown to wash the resin. As a next step, equilibrium buffer supplemented with 150 mM of imidazole (5 mL) was flown through the resin to elute His$_6$-tagged VlmD. Finally, the buffer solution of the eluted protein was changed to 50 mM potassium phosphate buffer (pH 7.5) by using Amicon Ultra-15 Centricon (Millipore, Beilerica, MA) with a pore size of 10 kDa. The concentration of the purified VlmD was measured by the Bio-Rad Protein Assay Kit (Bio-Rad, Hercules, CA) using bovine serum albumin (BSA) as a standard. For the in vitro enzyme assay, 50 mM potassium phosphate buffer (pH 7.5) was used, and supplemented with 1 mM of an amino acid precursor (i.e., glycine, L-alanine, L-2-aminobutyric acid, 2-aminoisobutyric acid, L-norvaline, L-valine, L-isovaline, L-norleucine, L-leucine, L-isoleucine, 1-aminocyclopentanecarboxylic acid, 1-aminocyclohexanecarboxylic acid, and L-2-phenylglycine), 0.1 mM of PLP, and 150 μg mL$^{-1}$ of the purified His$_6$-tagged VlmD. The enzymatic reaction was carried out for 2 h at 37 °C. All the in vitro enzyme assays were conducted in triplicates.

**Construction of bacterial strains and plasmids**. All bacterial strains and plasmids used in this study are listed in Supplementary Table 2. E. coli NEB 10-beta (New England Biolabs, Ipswich, MA) strain was used for gene cloning experiments. E. coli WL3110 strain[21], which is a lacI knockout mutant of E. coli W3110, was used as a host strain for the production of SCPAs. All the DNA manipulations were performed using standard procedures[38]. Cells were grown in Luria-Bertani (LB) broth or on LB plates (1.5%, w/v, agar) for the construction of plasmids and strains. When needed, antibiotics were added to the medium at the following concentrations: kanamycin (Km), 25 μg mL$^{-1}$; and ampicillin (Ap), 50 μg mL$^{-1}$.

All the primers used in this study were synthesized at Genotech (Daejeon, Korea), and are listed in Supplementary Table 3. The codon-optimized S. viridifaciens vlmD gene was synthesized at Genscript (Piscataway, NJ), and the codon-optimized G. stearothermophilus alaD gene was synthesized at Cosmogenetech (Seoul, Korea). DNA sequences of the codon-optimized genes are shown in Supplementary Table 4. Plasmid pKBRilvBNCED was obtained from the previous study[21]. The codon-optimized vlmD gene from S. viridifaciens was amplified by PCR using the primers vlmD_F1 and vlmD_R1. The amplified gene fragment was inserted into pTac15K at the EcoRI and KpnI sites to construct pTac15-vlmD$^{opt}$. To construct pET-his$_6$-vlmD$^{opt}$, the codon-optimized vlmD gene was amplified by PCR using the primers vlmD_F2 and vlmD_R2, and the resulting fragment was inserted into pET-22b(+) at the NdeI and EcoRI sites. To construct pTac15-vlmD$^{opt}$-alaD$^{opt}$, the codon-optimized G. stearothermophilus alaD gene was amplified using the primers alaD_F1 and alaD_R1. The amplified gene fragment was inserted into pTac15-vlmD$^{opt}$ at KpnI and PstI sites. To construct pTac15-alaD$^{opt}$, the codon-optimized G. stearothermophilus alaD gene was amplified using the primers alaD_F2 and alaD_R2, and the amplified gene fragment was inserted into pTac15K at the EcoRI site. The leuABCD gene fragment was generated by two sequential PCRs. The first PCR was conducted using the primers leuABCD_F1 and leuABCD_R1 with gDNA of E. coli W3110 as a template. The resulting amplified fragment was used as a template for the next PCR using the primers leuABCD_F2 and leuABCD_R2. This fragment was inserted into pTac15-vlmD$^{opt}$ at its PstI site to generate pTac15-vlmD$^{opt}$-pTac-leuABCD. Plasmid pTac15-leuABCD was constructed by site-directed mutagenesis of pTac15-vlmD$^{opt}$-pTac-leuABCD using inv_F1 and inv_R1 as primers.

The plasmids expressing synthetic sRNAs were constructed using Gibson assembly[39] as follows. The DNA fragment of plasmid backbone was prepared by inverse PCR with inv_F2 and inv_R2 primers using the plasmid pTac15-vlmD$^{opt}$ as a template. The DNA fragment of anti-ilvA expression cassette was prepared by PCR with anti-ilvA_F and anti-ilvA_R primers using the previously constructed plasmid[32] pWAS-anti-ilvA as a template. The DNA fragments of plasmid backbone and anti-ilvA expression cassette were ligated by Gibson assembly to construct pTac15-vlmD$^{opt}$-anti-ilvA. Similarly, pTac15-vlmD$^{opt}$-anti-ilvA-leuA was constructed by Gibson assembly of the DNA fragments of plasmid backbone (generated by inverse PCR with inv_F3 and inv_R3 primers using pTac15-vlmD$^{opt}$-anti-ilvA as a template) and anti-leuA expression cassette (generated by PCR with anti-leuA_F and anti-leuA_R primers using the previously constructed plasmid[32] pWAS-anti-leuA as a template). Lastly, pTac15-vlmD$^{opt}$-anti-ilvA-leuA-panB was constructed by Gibson assembly of the DNA fragment of plasmid backbone (generated by inverse PCR with inv_F4 and inv_R4 primers using pTac15-vlmD$^{opt}$-anti-ilvA-leuA as a template) and anti-panB expression cassette (generated by PCR with anti-panB_F and anti-panB_R primers using the previously constructed plasmid[32] pWAS-anti-panB as a template). Lastly, the plasmids pTac15-vlmD$^{opt}$-XYZ-alaD$^{opt}$ (X and Z represents original and replaced amino acid, respectively, in Y$^{th}$ residue of VlmD) were constructed by site-directed mutagenesis of pTac15-vlmD$^{opt}$-alaD$^{opt}$ using XYZ_F and XY_R as primers. For an example, pTac15-vlmD$^{opt}$-V94I-alaD$^{opt}$ was constructed by site-directed mutagenesis of pTac15-vlmD$^{opt}$-alaD$^{opt}$ using V94I_F and V94_R as primers.

**Cultivation condition**. The PA01 strain, E. coli WL3110 expressing the S. viridifaciens vlmD gene, was cultured in the MR medium (pH 7.0) supplemented with 10 g L$^{-1}$ of glucose and 2 g L$^{-1}$ of one of the 12 different amino acid precursors mentioned above. The EA01 strain producing ethylamine was cultured in the MR medium (pH 7.0) supplemented with 20 g L$^{-1}$ of glucose. The MR medium (pH 7.0) contains (per liter): 6.67 g KH$_2$PO$_4$, 4 g (NH$_4$)$_2$HPO$_4$, 0.8 g MgSO$_4$·7H$_2$O, 0.8 g citric acid, and 5 mL trace metal solution[40]. The trace metal solution contains (per liter of 0.5 M HCl): 10 g FeSO$_4$·7H$_2$O, 2 g CaCl$_2$, 2.2 g ZnSO$_4$·7H$_2$O, 0.5 g MnSO$_4$·4H$_2$O, 1 g CuSO$_4$·5H$_2$O, 0.1 g (NH$_4$)$_6$Mo$_7$O$_{24}$·4H$_2$O, and 0.02 g Na$_2$B$_4$O$_7$·10H$_2$O. Stock solutions containing glucose, MgSO$_4$·7H$_2$O, and one of the 12 different amino acid precursors were sterilized separately, and added to the MR medium later, before cell cultivation. Seed cultures were grown in a 25 mL test tube containing 10 mL of LB medium at 37 °C. After 8 h of cultivation, 1.5 mL of the seed culture was transferred to a 300 mL baffled flask containing 50 mL of culture medium, then cultivated at 37 °C and 200 rpm in a shaking incubator. The PA01 strain was cultured for 36 h, and the EA01 strain was cultured for 48 h. After cultivation, supernatants were collected for further analyses.

The iBA01 strain producing iso-butylamine was cultured in the NM2 medium previously reported[21]. The iAA01 strain producing iso-amlyamine was cultured in the NM2 medium without L-leucine. The NM2 medium contains per liter: 50 g glucose, 30 g CaCO$_3$, 12.5 g (NH$_4$)$_2$SO$_4$, 4.0 g KH$_2$PO$_4$, 2.0 g MgSO$_4$·7H$_2$O, 2 g yeast extract, 0.262 g L-isoleucine, 0.262 g L-leucine, 0.425 mg sodium D-pantothenate, and 5 mL trace metal solution. Seed cultures were prepared in the same manner as described above. After 8 h of cultivation, 1.5 mL of the seed culture was transferred to a 300 mL baffled flask containing 50 mL of culture medium, then cultivated for 48 h at 30 °C and 200 rpm in a shaking incubator. All the flask cultures were repeated three times.

Fed-batch fermentation of the iBA01 strain was performed in a 6.6 L bioreactor (Bioflo 3000; New Brunswick Scientific Co., Edison, NJ) containing 2 L of NM3 medium. The NM3 medium contains per liter: 20 g glucose, 20 g (NH$_4$)$_2$SO$_4$, 2 g KH$_2$PO$_4$, 0.4 g MgSO$_4$·7H$_2$O, 1.6 g NaCl, 5 g yeast extract, 0.262 g L-isoleucine, 0.262 g L-leucine, 0.362 mg sodium D-pantothenate, and 5 mL trace metal solution. Seed cultures were prepared by transferring 0.5 mL of culture broth (grown in 10 mL LB medium at 37 °C) into a 250 mL Erlenmeyer flask containing 50 mL LB medium, and cultured for 7 h at 30 °C in a shaking incubator at 200 rpm. The seed culture (200 mL)was inoculated to a bioreactor. Fermentation was performed at 30 °C, and the culture pH was adjusted to 7.0 by adding 28% (v/v) ammonia solution. Air was constantly flowed at 2 L min$^{-1}$. The dissolved oxygen (DO) level was kept at 40% of air saturation by automatically controlling the agitation speed from 200 to 1000 rpm and by supplying pure oxygen when the maximum agitation speed of 1000 rpm was reached. Feeding solution contained 400 g L$^{-1}$ of glucose, 20 g L$^{-1}$ of KH$_2$PO$_4$, 2.1 g L$^{-1}$ of L-isoleucine, 2.1 g L$^{-1}$ of L-leucine, and 3.62 mg L$^{-1}$ of sodium D-pantothenate. A total of 100 mL of the feeding solution was manually added when residual glucose concentration was less than 3 g L$^{-1}$. Fed-batch fermentation of the iBA02 strain was performed in the same condition as the iBA01 strain except that L-isoleucine, L-leucine, and sodium D-pantothenate were excluded in the medium and feeding solution.

**Analytical procedures**. Cell growth was monitored by measuring the optical density at 600 nm (OD$_{600}$) using Ultrospec 3000 spectrophotometer (Amersham Biosciences, Uppsala, Sweden). Glucose concentration was determined using the glucose analyzer (model 2700 STAT; Yellow Springs Instrument, Yellow Springs, OH). Concentrations of the 13 SCPAs (i.e., methylamine, ethylamine, n-propylamine, iso-propylamine, n-butylamine, iso-butylamine, (R)-sec-butylamine n-amylamine, iso-amylamine, 2-methylbutylamine, cyclopentylamine, cyclohexylamine, and benzylamine) were measured using high-performance liquid chromatography

(1100 Series HPLC; Agilent Technologies, Santa Clara, CA). Automatic precolumn derivatizations of the 13 SCPAs were performed by using o-phthaldialdehyde (OPA; Sigma, St. Louis, MO) in HPLC. The OPA derivatization reagent was prepared as follows[41,42]. The OPA reagent was prepared by dissolving 0.20 g of OPA in 9.0 mL of methanol, followed by adding 1.0 mL of 0.40 M (pH 9.0) borate buffer and 160 µL of 2-mercaptoethanol (a reducing reagent). All the culture broth samples, standard chemicals, OPA reagent, and borate buffer were filtered through a 0.2 µm PVDF syringe filter (FUTECS, Daejeon, Korea). For the derivatization, 1 µL of the sample was mixed with 5 µL of 0.40 M (pH 9.0) borate buffer. Following the addition of 1 µL of the OPA reagent, the mixture was injected into HPLC equipped with the Eclipse plus C18 column (4.6 × 150 mm; Agilent Technologies) operated at 25 °C. The mobile phase consists of solvent A (1.4 g $L^{-1}$ of $Na_2HPO_4$, 3.8 g $L^{-1}$ of $Na_2B_4O_7 \cdot 10H_2O$, and 8 mg $L^{-1}$ of $NaN_3$; adjusted to pH 7.2 with HCl) and solvent B (45% acetonitrile, 45% methanol, and 10% $H_2O$, in vol%), and was flown at 1.2 mL $min^{-1}$. The following gradients were applied: 0–0.5 min, 100% solvent A; 0.5–18 min, a linear gradient of B from 0 to 57%; 18–26 min, a linear gradient of solvent B from 57 to 100%; 26–29 min, 100% solvent B; 29-30 min, a linear gradient of solvent B from 100 to 0% (all in vol%). The derivatized SCPAs were detected by a variable wavelength detector (G1314A; Agilent Technologies) at 230 nm. The raw data were collected using Agilent ChemStation 4.03, and analyzed using Microsoft Excel 2013 (15.0.5111.1000) and OriginPro 2019.

For mass spectral analysis of the SCPAs, samples were derivatized with AccQ-Tag Ultra Derivatization Kit (Waters, Milford, MA) according to the manufacturer's protocol. Briefly, 10 µL of either in vitro assay sample or standard SCPA solution (100 mg $L^{-1}$) was mixed with 70 µL of AccQ-Tag Ultra borate buffer and 20 µL of AccQ-Tag reagent dissolved in 1.0 mL of AccQ-Tag Ultra reagent diluent. The derivatization reaction was carried out for 10 min at 55 °C. The derivatized samples or standard SCPA solutions were analyzed using HPLC (1100 Series HPLC; Agilent Technologies) connected with MS (LC/MSD VL; Agilent Technologies). The XBridge C18 column (5 µm, 4.6 × 150 mm; Waters) was used. The mobile phase comprises solvent C (50 mM ammonium acetate, adjusting pH to 9.3 by 28% $NH_4OH$ solution) and solvent D (acetonitrile). The mobile phase was flown at 0.4 mL $min^{-1}$. The following gradients were applied: 0-1 min, 90% C; 1–10 min, a linear gradient of D from 10 to 70%; 10–20 min, a linear gradient of D from 70 to 90%; 20–30 min, 90% D (all in vol%). The eluent was directed to MS using ESI positive ion mode under the following conditions: 100 V fragmentor, 12.0 L $min^{-1}$ drying gas flow, 350 °C drying gas temperature, 30 psig nebulizer pressure, and 3.0 kV capillary voltage. The scanned mass range was 50–500 m/z.

**Homology modeling and docking simulation of VlmD**. To obtain a homology model of VlmD, the native VlmD protein sequence was used as an input for the template-based modeling using SWISS-MODEL[43]. A homology model of VlmD was generated using GAD67 from *Homo sapiens* (PDB ID: 2OKJ). GAD67 was used as a template because it showed the greatest sequence identity (19.76%) with VlmD among proteins that use PLP as a cofactor. The PLP-L-alanine was subsequently docked into the active site of the resulting homology model of VlmD using AutoDock Vina (v1.1.2)[44]. Six target residues in the active site of the VlmD homology model were manually selected for mutations, which appeared to be interacting with L-alanine of PLP-L-alanine. To find candidate substitute residues for the six target residues, 250 homologous protein sequences were retrieved by aligning VlmD with protein sequences in the database of non-redundant protein sequences (nr) using BLASTP (v2.10.1). As a result, a total of 12 candidate substitute residues were identified from the homologous protein sequences detected. The six target residues from the active site of the VlmD homology model were experimentally modified to one of these 12 candidate substitute residues.

## Data availability

The data supporting the findings of this study are available within the article and its Supplementary Information and Supplementary Data. Additional data are available from the corresponding author upon reasonable request. The protein sequence of valine decarboxylase (VlmD) is available at UniProt-Q84F32 (https://www.uniprot.org/uniprot/Q84F32). The crystal structure of glutamic acid decarboxylase (GAD67) is available at PDB-2OKJ (http://www.rcsb.org/structure/2OKJ). Source data are provided with this paper. The engineered strains developed in this study can only be provided for non-commercial purposes as they are in commercial interest, and a patent is filed (Korean Patent Application No. 10-2020-0042756). Source data are provided with this paper.

## Code availability

Source code for the single-step retrobiosynthesis (Fig. 1b) and selection of precursors having a suitable molecular type (Fig. 1c) is available at https://bitbucket.org/kaistsystemsbiology/retro-precursor-selection.

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

## Acknowledgements

This work was supported by the Technology Development Program to Solve Climate Changes on Systems Metabolic Engineering for Biorefineries from the Ministry of Science and ICT through the National Research Foundation (NRF) of Korea (NRF-2012M1A2A2026556 and NRF-2012M1A2A2026557).

## Author contributions

S.Y.L. conceived the project. D.I.K. and T.U.C. performed experiments. D.I.K., H.U.K., and W.D.J. performed computational analyses. D.I.K., T.U.C., H.U.K., W.D.J., and S.Y.L. analyzed data. D.I.K., T.U.C., H.U.K., and S.Y.L. wrote the manuscript.

## Competing interests

The authors (D.I.K., T.U.C., and S.Y.L.) declare competing financial interests because the strains described in this paper are of commercial interest, and a patent is filed (Korean Patent Application No. 10-2020-0042756). All the authors (D.I.K., T.U.C., H.U.K., W.D.J., and S.Y.L.) declare no non-financial competing interests.
