## [Peer Review File · Nature Communications]

Reviewer #1 (Remarks to the Author):

This manuscript reports the enzymatic decarboxylation of ca. 12 different amino acids to yield the corresponding amine product in titres ranging from low (>100 mg per L) to moderate (ca. 1g per L). In all cases the key enzyme is valine decarboxylase (vlmD) which clearly shows a range of different activities with the different substrates. Unsurprisingly the best substrate is valine (yielding iso-butylamine) and the poorest substrates are those that are structurally quite different (e.g. phenyl glycine, alanine, norleucine etc.). The vlmD gene has been over-expressed in an E. coli strain which allows glucose to be used as a precursor for the production of these short-chain primary amines (SCPAs). This aspect of the paper is solid although seems to lack any real insight or major advance in understanding. I was surprised that the authors did not attempt protein engineering of vlmD in order to improve the activities towards the poor substrates.

The authors also report the use of computational tools for retrobiosynthesis in order to select candidate pathways for SCPA synthesis *in vivo*. Some of this information is displayed in Figure 4 which looks like a conventional metabolic map of known biosynthetic pathways involving amino acids, their precursors and possible products. It wasn't clear to me how the pathway that was ultimately chosen emerged from this retrobiosynthetic approach other than decarboxylation of amino acids is a fairly obvious and direct method for producing amines and has indeed been used many times in the past.

In the introduction the authors discuss conventional chemical approaches to SCPAs involving amination of alcohols at high temperature in the presence of metal catalysts. Although this chemistry is indeed quite dated it is very efficient and practiced on a very large scale under conditions of continuous manufacture to produce SCPAs at very low cost. The current methodology is a long way short of competing with that approach.

Reviewer #2 (Remarks to the Author):

The manuscript describes an interesting approach of using promiscuous enzymes for the production of a suite of SCPAs for production in E. coli. This work is of interest in the field of metabolic engineering, although fully utilizing promiscuous enzymes as a general platform may be more challenging to deploy for *in vivo* systems when endogenous metabolites may inadvertently react. This should be discussed more fully in the paper as a limitation. This issue notwithstanding, the paper has interesting results that will be of interest after changing some issues below:

A few of the sentences in this paper need revision for better flow (for example, avoid the duplication of words in the sentence such as lines 41-42)

I found the markings of “x” and “o” in the figures hard to follow and confusing to understand. A better nomenclature/demarcation will be necessary.

The predicted precursor analysis was hard to follow. How was this actually done? Insufficient details in the main text and methods section is provided. In some respects, the authors indicate the use of KEGG and other SMARTS systems, but then on lines 227 onward, the authors indicate that the ChemAxon Reactor CAN also be used. What was actually done here? This component is a critical aspect of the manuscript, yet the details are not well provided.

The in vitro reaction conditions were not well described. The authors indicate that there were His-tagged enzymes, but was this with a simple buffered solution or was this with cell free extract?

Reviewer #3 (Remarks to the Author):

Kim et al. developed a strategy to produce short chain primary amines using valine decarboxylase. This enzyme was known to decarboxylate branched-chain amino acids to their respective amines. Codon optimization of the valine decarboxylase gene *vlmD* from *Streptomyces viridifaciens* ensured good expression in *E. coli*. *VlmD* showed a broader substrate spectrum than previously thought and 12 amino acids were shown to be decarboxylated in an enzyme assay. Feeding experiments showed that 10 amino acids were decarboxylated in vivo by an *E. coli* strain producing *VlmD*. Importantly, three short chain primary amines were produced via fermentation from glucose by metabolically engineered *E. coli* strains, i.e. by strains previously engineered for amino acid production that now in addition expressed the *VlmD* encoding gene. The authors argue that decarboxylation by *VlmD* is irreversible and, thus, a driving force for metabolic flux. However, the titers for ethylamine (about 0.003 g/L), isobutylamine (about 1.1 g/L) and isoamylamine (about 0.09 g/L) are very low as compared to the titers of the precursor amino acids obtained with the parent strains that are expected to produce the amino acids L-alanine, L-leucine and L-valine with titers of 10-100 g/L. For comparison the authors should show amino acid production by the empty vector carrying parent strains with amine production by the *vlmD* expressing isogenic strain. The results obtained in vitro already indicated that decarboxylation of certain substrates was incomplete. Did degassing help?

Taken together, the authors present a new concept for production of short chain primary amines from glucose and ammonium in a simple medium by fermentation. It remains unclear why decarboxylation did not reach completion. Since the authors previously reported on production of diamines employing decarboxylases and reaching very high titers, production of one selected short chain primary amine to high titer, yield and productivity should be shown.

Author's Response to Reviewer #1:

This manuscript reports the enzymatic decarboxylation of ca. 12 different amino acids to yield the corresponding amine product in titres ranging from low (>100 mg per L) to moderate (ca. 1g per L). In all cases the key enzyme is valine decarboxylase (vlmD) which clearly shows a range of different activities with the different substrates. Unsurprisingly the best substrate is valine (yielding iso-butylamine) and the poorest substrates are those that are structurally quite different (e.g. phenyl glycine, alanine, norleucine etc.). The vlmD gene has been over-expressed in an E. coli strain which allows glucose to be used as a precursor for the production of these short-chain primary amines (SCPAs). This aspect of the paper is solid although seems to lack any real insight or major advance in understanding. I was surprised that the authors did not attempt protein engineering of vlmD in order to improve the activities towards the poor substrates.

[Response] We thank the reviewer for providing important and constructive comments. The reviewer's suggestion for additional experiments for improving the activity of VlmD toward poor substrates is well taken. Accordingly, we conducted protein engineering experiments for VlmD to improve its activity toward L-alanine, a precursor of ethylamine, which showed the lowest conversion efficiency among the 12 precursors. In short, a homology model of VlmD was constructed, and 12 target residues were identified via docking simulations for the homology model of VlmD. Among them, the ethylamine-producing EA01-V94I strain harboring the V94I mutant of VlmD showed the increased production titer (5.69 mg L⁻¹) of ethylamine. This value is 96.9% higher than 2.89 mg L⁻¹ from the EA01 strain possessing the native VlmD. These new results are added as a new section (pages 11 and 12 of the revised manuscript).

The authors also report the use of computational tools for retrobiosynthesis in order to select candidate pathways for SCPA synthesis in vivo. Some of this information is displayed in Figure 4 which looks like a conventional metabolic map of known biosynthetic pathways involving amino acids, their precursors and possible products. It wasn't clear to me how the pathway that was ultimately chosen emerged from this retrobiosynthetic approach other than decarboxylation of amino acids is a fairly obvious and direct method for producing amines and has indeed been used many times in the past.

[Response] A new Methods section on the precursor selection step (Fig. 1c,d) is added in the revised manuscript to make this point clear (pages 14 and 15 in the revised manuscript). For novel biosynthetic pathways designed for multiple SCPAs via the retrobiosynthesis and precursor selection step, there can be many enzymes that catalyze the corresponding reactions. In this study, we used VlmD as a proof of concept to implement the predicted biosynthetic reactions.

While it is true that the decarboxylation of amino acids is not uncommon in biological systems, it should be noted that the decarboxylation of amino acids had not been used in the past for the microbial production of SCPAs. Also, we newly characterized the activities of VlmD toward nine additional substrates in addition to the previously reported L-valine, L-leucine and L-isoleucine in this study.

In the introduction the authors discuss conventional chemical approaches to SCPAs involving amination of alcohols at high temperature in the presence of metal catalysts. Although this chemistry is indeed quite dated it is very efficient and practiced on a very large scale under conditions of continuous manufacture to produce SCPAs at very low cost. The current

methodology is a long way short of competing with that approach.

[Response] Thank you for the comment. As the reviewer commented, SCPAs can be efficiently produced via well-established chemical reactions. As in the cases of other bio-based chemicals, here we pursued the bio-based production of SCPAs in order to challenge the sustainable production of SCPAs from renewable resources as stated in the manuscript. Importantly, the strategy of retrobiosynthesis reported in this study can be generally extended to develop other platform strains for the production of a group of related chemicals beyond SCPAs.

Although optimization of production was not the objective of this study, subsequent metabolic engineering of a platform strain will allow more efficient production of a desired chemical, as we and other metabolic engineers have demonstrated for many different example chemicals over the years [recent reviews: Choi et al. Systems metabolic engineering strategies: integrating systems and synthetic biology with metabolic engineering, *Trends Biotechnol.*, 37(8): 817-837 (2019); Lee et al. A comprehensive metabolic map for production of bio-based chemicals, *Nature Catalysis*, 2: 18-33 (2019)].

Nonetheless, we performed additional experiments during the revision of our manuscript to demonstrate that the performance of the strain can be improved by further metabolic engineering. First, we conducted fed-batch fermentation of the *iso*-butylamine-producing iBA01 strain, which led to the production of 8.63 g L⁻¹ of *iso*-butylamine. This value corresponds to 6.50-fold increase, compared to the titer from the flask cultivation (1.15 g L⁻¹). Additional metabolic engineering was conducted on this strain to remove its auxotrophies toward L-isoleucine, L-leucine and D-pantothenate, which is not desirable for industrial fermentation. The resulting strain, iBA02, showed the further enhanced production titer of 10.67 g L⁻¹ *iso*-butylamine without supplementation of L-isoleucine, L-leucine and D-pantothenate in a medium. The results of these additional experiments are presented as a new section in the revised manuscript (pages 11 and 12, and Fig. 7 in the revised manuscript).

Author's Response to Reviewer # #2

The manuscript describes an interesting approach of using promiscuous enzymes for the production of a suite of SCPAs for production in *E. coli*. This work is of interest in the field of metabolic engineering, although fully utilizing promiscuous enzymes as a general platform may be more challenging to deploy for *in vivo* systems when endogenous metabolites may inadvertently react. This should be discussed more fully in the paper as a limitation. This issue notwithstanding, the paper has interesting results that will be of interest after changing some issues below:

[Response] We thank the reviewer for providing important and constructive comments. According to the reviewer's comment on the potential limitation of using a promiscuous enzyme *in vivo*, we newly inserted a relevant discussion by taking the iAA01 strain as an example (pages 9 and 10 in the revised manuscript). The iAA01 strain produced two byproducts, L-leucine and the *iso*-butylamine in addition to the target product *iso*-amylamine. In short, we noted that the production of *iso*-butylamine seems to have been caused by the unintended promiscuous activity of VImD toward L-valine, a precursor of *iso*-butylamine. This can be overcome by protein engineering for a particular product of interest as demonstrated through additional experiments in the revised manuscript. As already responded to the Reviewer #1's comment, we conducted protein engineering experiments for VImD to improve its activity toward L-alanine, a precursor of ethylamine, which showed the lowest conversion

efficiency among the 12 precursors. In short, a homology model of VlmD was constructed, and 12 target residues were identified via docking simulations for the homology model of VlmD. Among them, the ethylamine-producing EA01-V94I strain harboring the V94I mutant of VlmD showed the increased production titer (5.69 mg L^{-1}) of ethylamine. This value is 96.9% higher than 2.89 mg L^{-1} from the EA01 strain possessing the native VlmD. These new results are added as a new section (pages 11 and 12 of the revised manuscript).

A few of the sentences in this paper need revision for better flow (for example, avoid the duplication of words in the sentence such as lines 41-42)

[Response] Thank you. We carefully checked the entire manuscript.

I found the markings of “x” and “o” in the figures hard to follow and confusing to understand. A better nomenclature/demarcation will be necessary.

[Response] According to the reviewer’s suggestion, we used ‘check marks’ to indicate the selected precursors instead of using ‘X’ and ‘O’ (Figs. 1 and 2 of the revised manuscript).

The predicted precursor analysis was hard to follow. How was this actually done? Insufficient details in the main text and methods section is provided. In some respects, the authors indicate the use of KEGG and other SMARTS systems, but then on lines 227 onward, the authors indicate that the ChemAxon Reactor CAN also be used. What was actually done here? This component is a critical aspect of the manuscript, yet the details are not well provided.

[Response] Sorry for not clearly providing the relevant information, and thank you for pointing this out. By ‘precursor analysis’, we believe the reviewer is referring to ‘precursor selection step (Fig. 1c,d). According to the reviewer’s great suggestion, a new section is prepared in the Methods of the revised manuscript, which describes the selection procedure of precursors (pages 14 and 15 in the revised manuscript). Also, we mentioned ‘ChemAxon Reactor’ in Methods, although we did not use it in this study, for the people who wish to use our reaction rules or reproduce our results using ChemAxon Reactor.

The in vitro reaction conditions were not well described. The authors indicate that there were His-tagged enzymes, but was this with a simple buffered solution or was this with cell free extract?

[Response] Again, thank you for the comment. We used the ‘purified’ His₆-tagged VlmD that is with a simple buffer solution. This point is clarified in Methods of the revised manuscript (page 16).

Author’s Response to Reviewer # #3:

Kim et al. developed a strategy to produce short chain primary amines using valine decarboxylase. This enzyme was known to decarboxylate branched-chain amino acids to their respective amines. Codon optimization of the valine decarboxylase gene *vlmD* from *Streptomyces viridifaciens* ensured good expression in *E. coli*. VlmD showed a broader substrate spectrum than previously thought and 12 amino acids were shown to be decarboxylated in an enzyme assay. Feeding experiments showed that 10 amino acids were decarboxylated in vivo by an *E. coli* strain producing VlmD. Importantly, three short chain primary amines were produced via fermentation from glucose by metabolically engineered *E.*

coli strains, i.e. by strains previously engineered for amino acid production that now in addition expressed the VImD encoding gene.

[Response] We thank the reviewer for the important and constructive comments.

The authors argue that decarboxylation by VImD is irreversible and, thus, a driving force for metabolic flux. However, the titers for ethylamine (about 0.003 g/L), isobutylamine (about 1.1 g/L) and isoamylamine (about 0.09 g/L) are very low as compared to the titers of the precursor amino acids obtained with the parent strains that are expected to produce the amino acids L-alanine, L-leucine and L-valine with titers of 10-100 g/L. For comparison the authors should show amino acid production by the empty vector carrying parent strains with amine production by the vImD expressing isogenic strain.

[Response] Thank you for the great comment. As the reviewer advised, we measured the concentrations of amino acid precursors from flask cultures of control (parental) strains, having an empty plasmid, and their counterpart SCPAs-producing strains for ethylamine, *iso*-butylamine and *iso*-amylamine (pages 8, 9, Fig. 6 and Supplementary Fig. 4 of the revised manuscript). According to the newly prepared data, VImD showed a strong driving force toward *iso*-butylamine because the *iso*-butylamine titer achieved by the iBA01 strain (1154.74 mg L⁻¹; equivalent to 15.79 mM) was higher than L-valine titer obtained with the control strain NC02 having an empty plasmid (524.91 mg L⁻¹; equivalent to 4.48 mM) (Fig. 6b in the revised manuscript). No L-valine was accumulated as a byproduct in the case of the iBA01 strain. In the cases of ethylamine and *iso*-amylamine, their production titers were substantially lower than the titers of their precursors (L-alanine and L-leucine, respectively) obtained with the counterpart control strains due to the relatively lower activities of VImD on these substrates compared with L-valine (Fig. 6a,c in the revised manuscript).

The results obtained in vitro already indicated that decarboxylation of certain substrates was incomplete. Did degassing help? Taken together, the authors present a new concept for production of short chain primary amines from glucose and ammonium in a simple medium by fermentation. It remains unclear why decarboxylation did not reach completion.

[Response] Thank you for the great comment. The production titers of precursors from the control strains having an empty plasmid suggest that the suboptimal activity of VImD seems to be a main reason for the incomplete conversion of L-alanine to ethylamine and L-leucine to *iso*-amylamine. A balance should be made between metabolic fluxes that contribute to the generation and conversion of a precursor. In case of *iso*-amylamine, as in our response for the Reviewer #2's comment, the generation of *iso*-butylamine as a byproduct seems to have been caused by the promiscuous activity of VImD toward L-valine (pages 9 and 10 in the revised manuscript).

Degassing might be useful for enzymatic conversion (as shown in in vitro assay in our study) for *iso*-butylamine, toward which VImD showed a strong driving force as mentioned above (Fig. 6b). In contrast, the effects of the CO₂ degassing seemed to be less obvious for ethylamine and *iso*-amylamine because the conversion efficiencies of L-alanine and L-leucine into these SCPAs were too low (Fig. 6a,c in the revised manuscript). In real fermentation of engineered strains, however, degassing should be carefully examined as CO₂ generated from the SCPA-producing strains was removed by aeration, but CO₂ is also needed for carboxylation reactions in the cell.

Since the authors previously reported on production of diamines employing decarboxylases and reaching very high titers, production of one selected short chain primary amine to high titer,

yield and productivity should be shown.

[Response] Thank you. We intended to focus on the development of a general 'retrobiosynthesis' strategy in the paper, and thus overproduction studies are beyond the scope of this paper. However, as the reviewer's point is constructive, we conducted further metabolic engineering of *iso*-butylamine producing strain by removing auxotrophies via sRNA-based gene modulation and performed fed-batch fermentations. The production titer and productivity of *iso*-butylamine obtained were 10.67 g L⁻¹ and 0.36 g L⁻¹ h⁻¹, respectively. These new results are presented as a new section in the revised manuscript (pages 10 and 11 of the revised manuscript).

Also, according to the Reviewer #1's first comment, we conducted protein engineering for VImD where the ethylamine-producing EA01-V94I strain harboring the V94I mutant of VImD showed the increased production titer of ethylamine, 5.69 mg L⁻¹, which corresponds to 96.9% increase, compared with the EA01 strain having the native VImD (pages 11 and 12 of the revised manuscript).

Reviewer #1 (Remarks to the Author):

I am satisfied that the authors have now addressed the reviewers' comments.

Reviewer #2 (Remarks to the Author):

The authors have nicely addressed the comments from the prior review and their additional explanations have greatly improved the manuscript.

Reviewer #3 (Remarks to the Author):

The authors have performed additional experiments to address my concerns. The results obtained supported their conclusions. Thus, the authors have convincingly responded to my criticism.